# Increasing secular trends in height and obesity in children with type 1 diabetes: JSGIT cohort

Mie Mochizuki[1]*, Yoshiya Ito[2], Hiroshi Yokomichi[3], Toru Kikuchi[4], Shun Soneda[5], Ikuma Musha[4], Makoto Anzou[6], Koji Kobayashi[1], Kumihiro Matsuo[7], Shigetaka Sugihara[8], Nozomu Sasaki[4], Nobuo Matsuura[9], Shin Amemiya[4], On behalf of The Japanese Study Group of Insulin Therapy for Childhood and Adolescent Diabetes (JSGIT)[¶]

1 Department of Pediatrics, University of Yamanashi, Chuo, Japan, 2 Japanese Red Cross Hokkaido College of Nursing, Kitami, Japan, 3 Department of Health Sciences, University of Yamanashi, Chuo, Japan, 4 Department of Pediatrics, Saitama Medical University, Iruma, Japan, 5 Department of Pediatrics, St. Marianna University School of Medicine, Kawasaki, Japan, 6 Department of Pediatrics, Tokyo Metropolitan Ohtsuka Hospital, Tokyo, Japan, 7 Department of Pediatrics, Asahikawa Medical University, Asahikawa, Japan, 8 Department of Pediatrics, Tokyo Women's Medical University Medical Center East, Tokyo, Japan, 9 Department of Pediatrics, Bibai City Hospital, Bibai, Japan

¶ Membership of The Japanese Study Group of Insulin Therapy for Childhood and Adolescent Diabetes is listed in the Acknowledgments.
* mmie@sweet.ocn.ne.jp

**Data Availability Statement:** Data cannot be shared publicly because data contain sensitive patient information. Data are available on request from the Japanese Study Group of Insulin Therapy for Childhood and Adolescent Diabetes and Ethics

## Abstract

### Background

Recently, anthropometric indices in children with type 1 diabetes mellitus (T1DM) have begun to change.

### Objective

To examine secular trends in patients' anthropometric indices.

### Subjects

Japanese children with T1DM from the 1995, 2000, 2008 and 2013 cohorts of The Japanese Study Group of Insulin Therapy for Childhood and Adolescent Diabetes.

### Methods

We analysed serum haemoglobin A1c (HbA1c) levels, the incidence of severe hypoglycaemic events, the types and doses of insulin, height standard deviation scores (SDS), body mass index (BMI) percentiles compared with healthy Japanese children and obesity prevalence over time. We also stratified the patients according to glycaemic control levels of <58 mmol/mol (optimal), 58–75 mmol/mol (suboptimal) and ≥75 mmol/mol (high-risk).

### Results

Data for 513–978 patients from each of the cohorts were analysed. The incidence of severe hypoglycaemic events decreased over time (from 21 to 4.8/100 patient-years), while the

Committee of Saitama Medical University (contact via rinri@saitama-med.ac.jp) for researchers who meet the criteria for access to confidential data.

**Funding:** This work was supported by Grants-in-Aid for Scientific Research from the Ministry of Education, Culture, Sports, Science and Technology of Japan (grant number: KAKENHI JP19K10658 to M. Mochizuki, JP18K17376 to H. Yokomichi, and JP20K11653 to T. Kikuchi), Health Labour Sciences Research Grant from the Ministry of Health, Labour and Welfare of Japan (grant number: H29-Junkan-Ippan-004 to T. Kikuchi), and research aid from the Japan Diabetes Foundation to T. Kikuchi and the Kawano Masanori Memorial Public Interest Incorporated Foundation for Promotion of Pediatrics to T. Kikuchi. The funders had no role in study design, data collection, and analysis, decision to publish, or preparation of the manuscript.

**Competing interests:** The authors have declared that no competing interests exist.

proportion of insulin analogue doses increased (14.6% to 98.6%). In addition, patient height SDS (−0.22 to +0.17), BMI percentile (52.1 to 58.7) and obesity prevalence (2.1% to 5.1%) increased. Height SDS increased in all of the glycaemic control subgroups, while BMI percentile and obesity prevalence increased in the suboptimal and high-risk groups.

## Conclusions

Since 1995, the average height of children with T1DM has increased in parallel with increasing insulin doses. Clinicians should be aware of increased BMI in these patients and the associated risk of developing cardiovascular disease in the future.

## Introduction

Recently, clinicians have noticed improvements in glycaemic control in patients with type 1 diabetes mellitus (T1DM). We previously reported a secular trend in glycaemic control improvement in Japanese patients [1]. Although glycaemic control varies among countries, age groups and patients [2], recent studies have reported a decreased mean haemoglobin A1c (HbA1c) level in Japanese patients with T1DM of between 7.7% and 8.2% [1, 3, 4]. The 2007 International Society for Pediatric and Adolescent Diabetes (ISPAD) consensus guidelines management objectives indicated that unless there was severe hypoglycaemia and repeated hypoglycaemia, the target HbA1c for all children under the age of 18 was < 7.5% [5]. The Japanese Society for Pediatric Endocrinology (JSPE) translated these guidelines into Japanese in 2008 [6]. The JSPE and the Japan Diabetes Society (JDS) recommended an HbA1c levels of < 7.5% for optimal glycaemic control since 2011, consistent with the ISPAD recommendations [7]. The Japanese Study Group of Insulin Therapy for Childhood and Adolescent Diabetes (JSGIT) uses this target in Japanese children and adolescents with T1DM.

Given these new, lower target HbA1c levels, the major focus of T1DM treatment has shifted from decreasing the incidence of serum hypoglycaemic levels to preventing diabetic complications. These complications include hypoglycaemia, vascular complications and obesity [8]. Disrupted growth and maturation are also critical complications for paediatric patients. Given the improvement in glycaemic control over time, paediatric diabetologists should be aware of secular changes in the growth and maturation of their patients.

In 2010, researchers in the USA reported temporal patterns in overweight and obesity in adult patients with T1DM, and suggested that the prevalence of overweight was increasing in this population [9]. However, little information is available regarding anthropometric measurements in paediatric patients, and, in particular, in Japanese paediatric patients with T1DM. In this study, we analysed changes in height and body mass index (BMI) of cohorts of Japanese paediatric patients with T1DM. We also assessed changes in the prevalence of overweight and obesity, which could increase the risk of future cardiovascular complications.

## Materials and methods

### Participants

Data for paediatric patients with T1DM were obtained from the JSGIT cohorts [1, 10]. The JSGIT is the only multi-institutional joint research group in Japan, and the group was established in 1995 for patients with childhood-onset type 1 diabetes [10]. The first cohort study began in 1995 and was designed to improve the treatment and glycaemic control of diabetic

Table 1. Participants' characteristics.

|  | 1995 | 2000 | 2008 | 2013 | *P*-value for the trend* |
|---|---|---|---|---|---|
| Number of Institutions | 37 | 51 | 64 | 68 | |
| Number of participants (% boys) | 513 (41.7) | 685 (37.4) | 734 (39.9) | 978 (41.3) | <0.001(0.632) |
| Age, years | 13.4 (2.9) | 13.1 (3.7) | 12.4 (3.6) | 11.9 (3.6) | <0.001 |
| Age at onset, years | 7.2 (3.7) | 7.1 (3.8) | 6.6 (3.7) | 6.7 (3.7) | <0.001 |
| Duration of diabetes, years | 6.2 (3.5) | 5.9 (3.5) | 5.8 (3.5) | 5.1 (3.6) | <0.001 |

The data are presented as mean (standard deviation) or number (%).

*$P$-values for the trends as determined by Cochran–Armitage or Jonckheere–Terpstra tests.

children in Japan. The number of participating institutions and registered patients is increasing cohort by cohort. The current study was a cross-sectional study performed using data from the entry year for the four cohort studies. We analysed patients' baseline data from cohorts recruited in 1995, 2000, 2008 and 2013, and we excluded data from patients who had T1DM for less than 6 months. Patients were diagnosed with T1DM by their paediatricians according to the criteria established by the JDS and the American Diabetes Association [11, 12]. Participants' clinical characteristic are shown in Table 1.

## Anthropometric indices

We recorded the height, weight and HbA1c levels for all of the patients, and we analysed data collected between July and October for each cohort. BMI was calculated as weight in kilograms divided by the square of the height in metres. We defined overweight and obesity as BMI in the 85th–94th percentile and BMI $\geq$ 95th percentile [13], respectively, according to the 2000 JSPE data for each age and sex [14]. We also evaluated BMI standard deviation scores (SDSs) and height SDSs to compare the data from patients across all age groups [15]. SDS values for controls were calculated using data from healthy Japanese children recorded in the National Survey in 2000 [16].

## Other parameters

According to the ISPAD guidelines [17], we categorised patients into three groups according to their level of glycaemic control as follows: optimal: < 58 mmol/mol using Independent Federal Constitution Commission units (IFCC) (< 7.5% in the National Glycohemoglobin Standardization Program (NGSP) guidelines); suboptimal: 58–75 mmol/mol (7.5%–9.0%); and high-risk: $\geq$ 75 mmol/mol ($\geq$ 9.0%). We also recorded the incidence of severe hypoglycaemic events, as well as the types and doses of insulin formulations used. Severe hypoglycaemia was defined as an event associated with severe cognitive impairment, including coma and convulsions, requiring external assistance by another person [18]. We categorised patients into four age groups; 5–6-year-old group, 10–11-year-old group, 12–13-year-old group and 15–16-year-old group, according to sex. Pubertal stage was classified as follows: in boys, the 5–6-year-old and 10–11-year-old groups indicated the pre-pubertal stage; the 12–13-year-old group indicated the early pubertal stage and the 15–16-year-old group indicted the middle to the end of the pubertal stage. In girls, the 5–6-year-old group indicated the pre-pubertal stage; the 10–11-year-old group indicated the early pubertal stage; the 12–13-year-old group indicated the middle pubertal stage and the 15–16-year-old group indicated the end of the pubertal stage [19].

## Statistical analysis

We described glycaemic control, the incidence of severe hypoglycaemic events, types and doses of insulin, the proportion of patients using bolus insulin at tea-time, height SDSs, BMI and percentile and the incidence of obesity and overweight over time. We also analysed secular change in these indices over time according to glycaemic control groups. Secular change in height SDSs and BMI percentile among boys and girls were analysed over time according to the four age groups: 5–6-year-old group, 10–11-year-old group, 12–13-year-old group and 15–16-year-old group.

We used the Mann–Whitney U test to compare skewed data and the chi-square test to compare categorical data. We statistically evaluated secular trends by the Jonckheere–Terpstra test for continuous variables and the Cochran–Armitage test for categorical variables. We reported descriptive statistics as means and standard deviations (SDs). Two-sided probability values less than 0.05 were considered statistically significant. All statistical tests and descriptive analyses were performed using JMP software (version 13.0; SAS Institute Inc., Cary, NC, USA) and SAS statistical software (version 9.3; SAS Institute Inc., Cary, NC, USA).

## Ethics approval statement

This study was approved by the institutional review boards of the ethics committee of Saitama Medical University (no: 12–077) and School of Medicine, University of Yamanashi (no: 1817), and the study was conducted in accordance with the provisions of the Declaration of Helsinki. Written consent was obtained from all patients or their caregivers.

## Results

In total, 513, 685, 734 and 978 patients from 1995, 2000, 2008 and 2013, respectively, were included in the study. The percentage of boys in these cohorts was 41.7%, 37.4%, 39.9% and 41.3%, respectively, and the mean disease duration at baseline was 5–6 years (Tables 1 and 2). Table 2 shows the mean HbA1c levels, dose and type of insulin, incidence of severe hypoglycaemic events, SDSs for height and BMI, BMI percentile and prevalence of obesity in the patients with T1DM selected for inclusion (Fig 1). Over time, there were statistically significant decreases in HbA1c levels and the incidence of severe hypoglycaemic events and statistically significant increases in the proportion of patients who took bolus insulin at tea-time, as well as in height SDSs, BMI SDSs and the prevalence of overweight and obesity.

Table 3 and Fig 2 show the secular changes in the means of height SDS, BMI percentile and the prevalence of obesity compared with the glycaemic control group. In all glycaemic control groups, height SDSs showed an increasing trend over time. In the suboptimal and high-risk glycaemic control groups, the BMI percentile and prevalence of obesity showed increasing trends. Total daily insulin dose showed an increasing trend in the order of high-risk control group, suboptimal control group and optimal control group (Fig 2). Height SDSs, BMI percentile and proportion of obesity between T1DM with or without using bolus insulin at tea-time did not differ significantly (−0.20 (1.23) vs. −0.42 (1.80), 58.1 (25.8) vs. 58.3 (24.5) and 0.0% vs. 1.8% in 2000; 0.04 (1.07) vs. 0.06 (1.09), 58.6 (25.8) vs. 56.5 (26.7) and 18.4% vs. 20.1% in 2008; 0.17 (1.07) vs. 0.17 (0.89), 59.2 (26.3) vs. 57.4 (25.5) and 28.0% vs. 26.5% in 2013, respectively).

The proportions of girls were higher in each cohort, but there were no significant differences between the cohorts. A dominant proportion of girls in a cohort is a common characteristic in Japanese patients with T1DM [1].

According to the age groups, height SDSs and BMI percentile showed significant increasing trends in the 15–16-year-old group ($P = 0.0003$ and $P = 0.014$ in boys and $P < 0.0001$ and $P = 0.0055$ in girls, respectively) (Fig 3).

**Table 2. Secular changes in height SDSs and the prevalence of overweight and obesity in Japanese children with type 1 diabetes mellitus.**

| Cohort | 1995 | 2000 | 2008 | 2013 | P-value for the trend* |
|---|---|---|---|---|---|
| Number, (% male) | 513 (41.7) | 685 (37.4) | 734 (39.9) | 978 (41.3) | |
| HbA1c, % (SD) | 9.4 (2.0) | 8.4 (1.6) | 7.8 (1.1) | 8.0 (1.2) | <0.001 |
| HbA1c, mmol/mol (SD) | 78.9 (22.4) | 68.3 (18.0) | 61.4 (12.5) | 64.1 (13.3) | <0.001 |
| Incidence of severe hypoglycaemic events per 100 patient-years (95% CI) | No data | 21.0 (8.5–33.6) | 7.4 (3.1–11.6) | 4.8 (3.2–6.3) | <0.001 |
| Proportion of patients using insulin analogues, % | 0 | 14.6 | 94.7 | 98.6 | <0.001 |
| As bolus, % | 0 | 14.6 | 88.6 | 87.2 | <0.001 |
| As basal, % | 0 | 0 | 94.7 | 96.5 | <0.001 |
| Proportion of patients using bolus insulin at tea-time, % | No data | 1.72 | 19.7 | 26.3 | <0.001 |
| Total daily insulin dose, units/kg/day (SD) | 1.01 (0.32) | 1.08 (0.35)† | 1.09 (0.33)‡ | 1.02 (0.37) | 0.992 |
| Height SDS, mean (SD) | −0.22 (1.07) | −0.21 (1.24) | +0.04 (1.08) | +0.17 (1.02) | <0.001 |
| BMI >85th percentile, (%) | 59 (11.50) | 114 (16.64) | 136 (18.53) | 196 (20.04) | <0.001 |
| BMI percentile, mean (SD) | 52.1 (25.3) | 58.1 (25.8) | 58.2 (26.0) | 58.7 (26.1) | <0.001 |
| BMI SDS, mean (SD) | +0.02 (1.18) | +0.21 (1.02) | +0.25 (0.93) | +0.28 (0.90) | <0.001 |
| Obese (%) | 11 (2.14) | 25 (3.65) | 48 (5.18) | 50 (5.11) | 0.004 |
| Overweight (%) | 48 (9.36) | 89 (12.99) | 98 (13.35) | 146 (14.93) | 0.005 |

The data are presented as mean (SD) or number (%).

*P-values for the trends as determined by the Cochran–Armitage or Jonckheere–Terpstra tests using data from 1995 to 2013.

†P = 0.001

‡P<0.001 vs. 1995 as determined by the Mann–Whitney U test.

SD, standard deviation; 95% CI, 95% confidence interval; BMI, body mass index; SDS, standard deviation score; obese, BMI ≥95th percentile; overweight, BMI in the 85th–94th percentile; 0, no patients used insulin analogues or insulin analogues as a bolus or as basal therapy.

## Discussion

Our data suggest that height SDSs, BMI percentile and the prevalence of obesity have increased in Japanese paediatric patients with T1DM from 1995 to 2013. The height SDSs of these patients are currently similar to those of healthy Japanese children. These trends were accompanied by an increase in the use of insulin analogues and the use of bolus insulin at tea-time. A trend toward increased BMI percentile and obesity prevalence was observed in the suboptimal and high-risk glycaemic control groups.

Paediatric patients with T1DM have been reported to be shorter than healthy children even after initiating insulin therapy, and this phenomenon is particularly pronounced in patients with poor glycaemic control [20, 21]. In patients with T1DM, peripheral insulin administration results in lower levels of portal insulin, which leads to higher concentrations of growth hormone binding proteins (GHBPs) than in healthy individuals [22]. Higher levels of GHBPs, in turn, lower free insulin-like growth hormone-1 (IGF-1) concentrations [22]. Thus, the impaired growth seen in children with T1DM may be due to low portal insulin levels [22]. In our study, we observed increased height SDSs over time in all of the cohorts, as shown in Table 2. This suggests that, especially in the high-risk glycaemic control group, height can be increased by peripherally administering larger doses of insulin, which lead to higher levels of portal insulin.

### Comparison of the trends in the cohorts with those in the general population

The proportion of overweight children (weight > 20% over the standard weight adjusted for sex and height) in the Japanese general population increased from 1995 to 2003, but did not

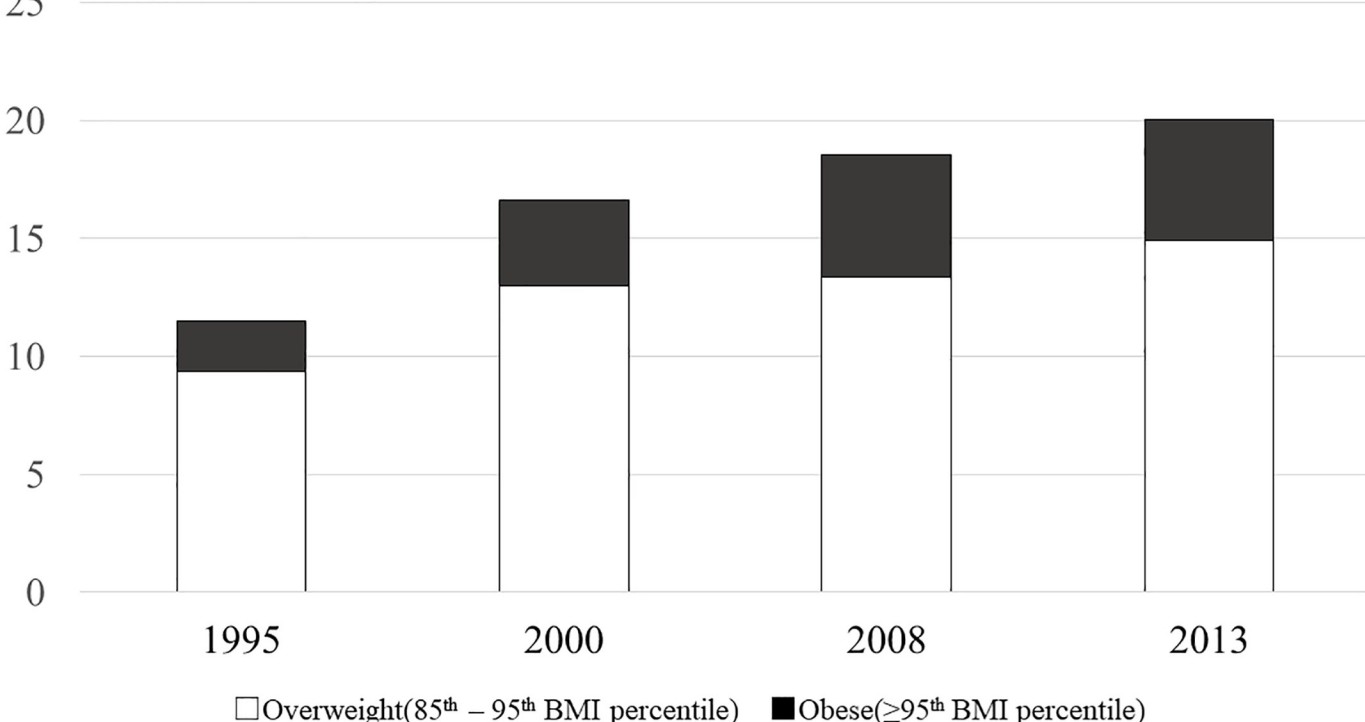

**Fig 1. Prevalence of obesity and overweight in each cohort.** The black bars indicate the prevalence of obesity ($\geq$ 95[th] BMI percentile), and the white bars indicate overweight (85[th]–95[th] BMI percentile).

change from 2003 to 2013 [23]. The proportion of overweight and obese children with T1DM in this study's cohorts increased. The difference in the proportion of obesity from 2008 to 2013 might be associated with having diabetes.

**Table 3. Secular changes in height SDSs and BMI percentile according to the level of glycaemic control in Japanese children with type 1 diabetes mellitus.**

| Cohort | 1995 | 2000 | 2008 | 2013 | P-value for the trend* |
|---|---|---|---|---|---|
| | (n = 513) | (n = 685) | (n = 734) | (n = 978) | |
| **Mean height SDS** | | | | | |
| HbA1c <58 mmol/mol (7.5%) | −0.03 | −0.13 | +0.13 | +0.03 | <0.001 |
| HbA1c 58–75 mmol/mol (7.5%–8.9%) | −0.12 | −0.17 | +0.08 | −0.08 | <0.001 |
| HbA1c ≥75 mmol/mol (9.0%) | −0.35 | −0.18 | −0.32 | −0.13 | <0.001 |
| **Mean BMI percentile** | | | | | |
| HbA1c <58 mmol/mol (7.5%) | 52.4 | 55.9 | 57.2 | 57.9 | 0.0530 |
| HbA1c 58–75 mmol/mol (7.5%–8.9%) | 53.0 | 59.5 | 59.6 | 60.0 | 0.0100 |
| HbA1c ≥75 mmol/mol (9.0%) | 51.8 | 59.5 | 58.3 | 58.4 | 0.0020 |
| **Proportion of obesity (%)** | | | | | |
| HbA1c <58 mmol/mol (7.5%) | 2.3 | 3.8 | 4.1 | 5.3 | 0.110 |
| HbA1c 58–75 mmol/mol (7.5%–8.9%) | 1.7 | 2.8 | 5.1 | 4.4 | 0.011 |
| HbA1c ≥75 mmol/mol (9.0%) | 2.7 | 4.9 | 9.6 | 7.0 | 0.021 |

*P-value for the trend as determined by the Cochran–Armitage or Jonckheere–Terpstra tests.

HbA1c, haemoglobin A1c; BMI, body mass index; SDS, standard deviation score; obesity, BMI ≥95[th] percentile.

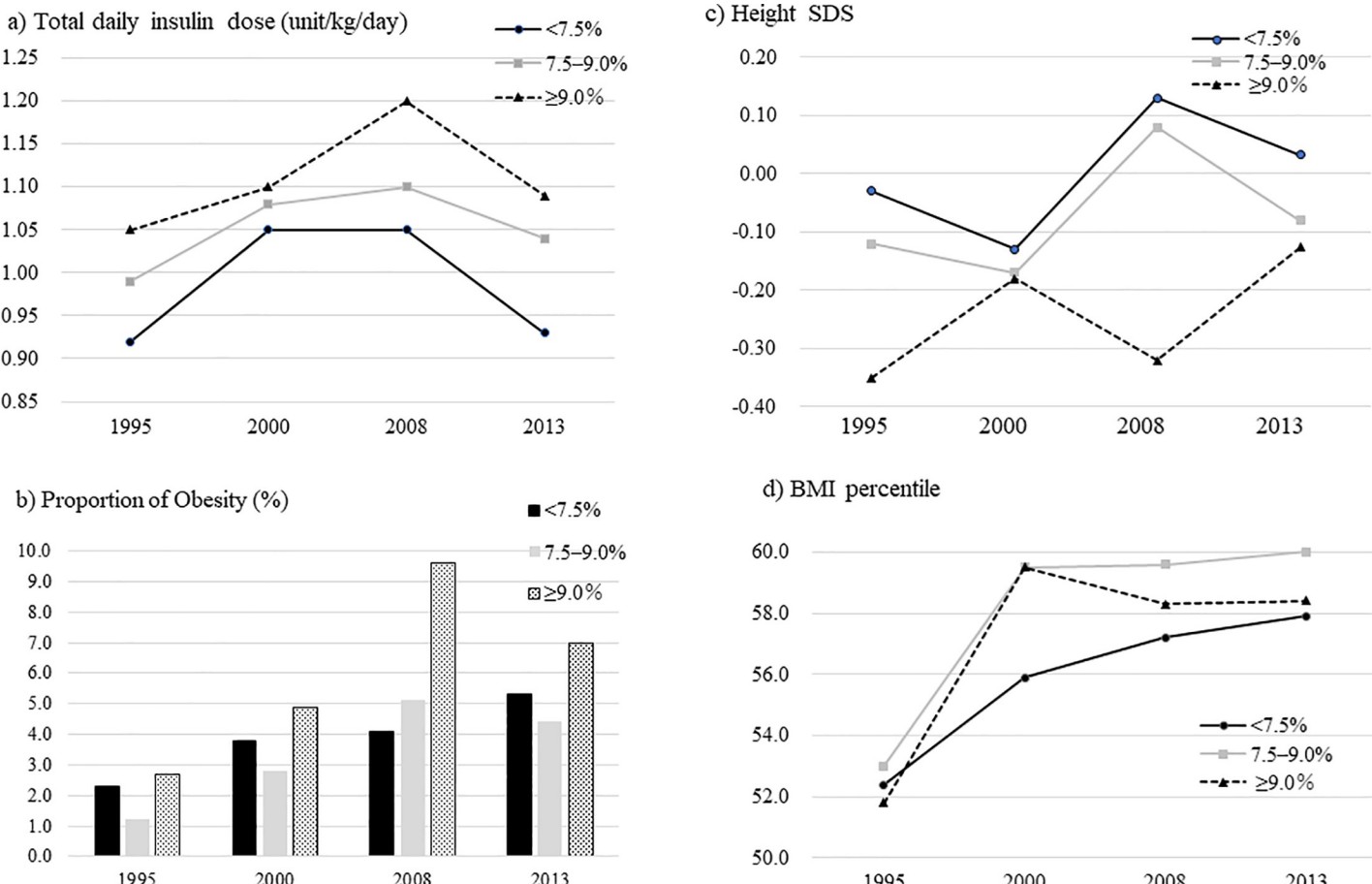

**Fig 2. Secular changes in total daily insulin dose, proportion of obesity, height SDSs and BMI percentile according to the level of glycaemic control in Japanese children with type 1 diabetes mellitus.** a) Mean total daily insulin dose (units/kg/day), b) Prevalence of obesity (%), c) Mean height SDS and d) Mean BMI percentile. The black lines and bars indicate the optimal control group, the grey lines and bars indicate the sub-optimal control group and the broken lines and dotted bars indicate the high-risk control group. BMI, body mass index; SDS, standard deviation score.

Considering puberty, the onset of puberty among Japanese children advanced from 1950 to 2010, but stabilised by 1970 [19]. Height SDSs, BMI percentile and the proportion of overweight and obese children with T1DM could not be evaluated using Japanese data for the year 2000.

Height SDSs and BMI percentile showed a significant increasing trend in the 15–16-year-old group for both boys and girls in the cohorts, which reflect increased height SDSs and BMI percentiles at the end of the pubertal stage.

## Shift to analogue insulins and increased total daily dose

The proportion of obese patients increased in the suboptimal and high-risk glycaemic control groups over time. Previous studies also reported an increase in BMI in paediatric patients with T1DM compared with healthy children [9]. The increased use of insulin analogues, and a shift from NPH insulin to long-acting insulin and from regular insulin to rapid-acting insulin analogues, which is associated with a decreased risk of hypoglycaemic events, may account for the increase in BMI that we observed in our study [24]. Although data regarding dietary intake were not available for the cohorts analysed in our study, we suspect that the high proportion of obesity observed in the suboptimal and high-risk glycaemic control groups may be due to excessive food intake and inappropriate insulin administration. Increased body fat in these

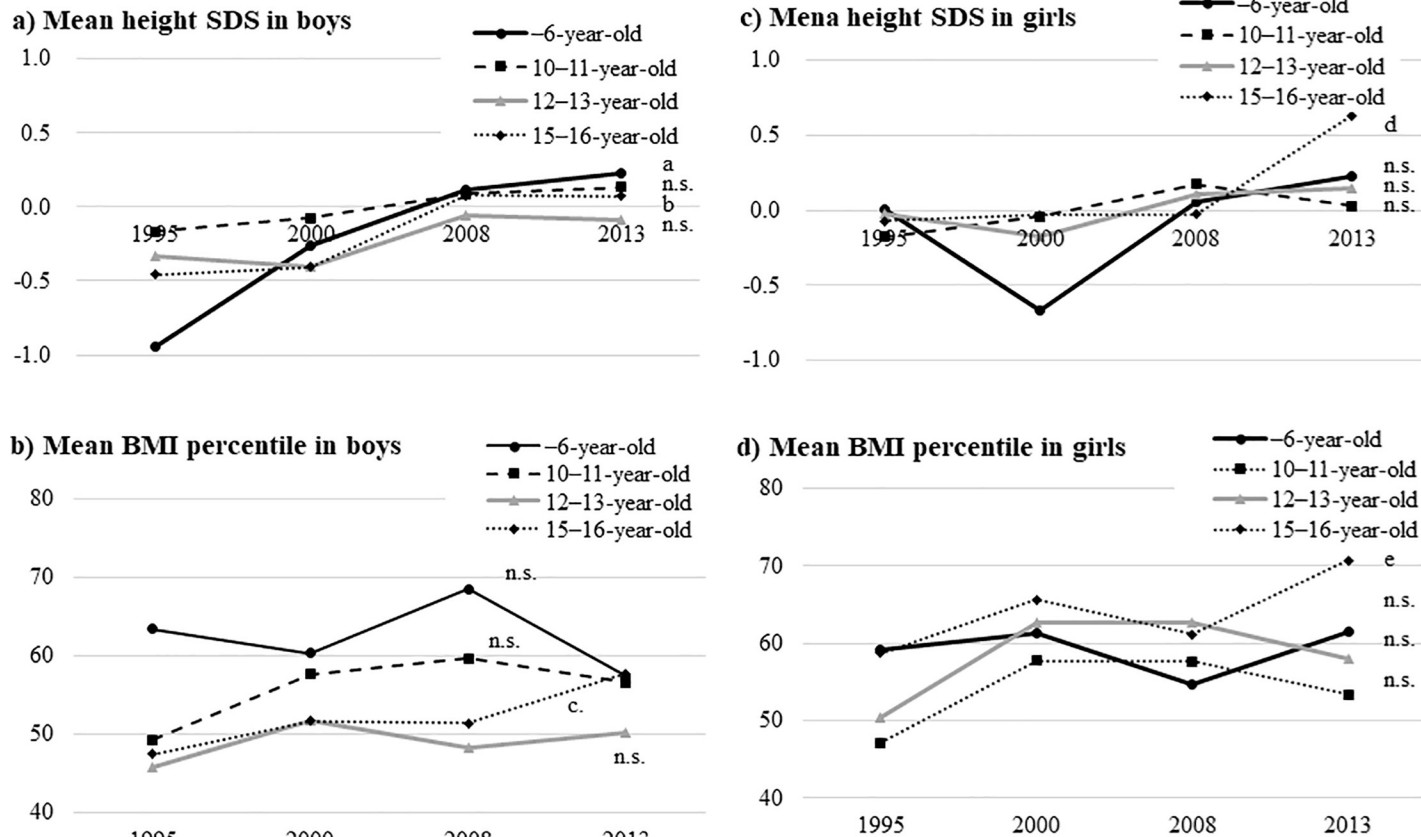

**Fig 3. Secular changes in height SDSs and BMI percentile in the four age groups.** a) Mean height SDS in boys, b) Mean BMI percentile in boys, c) Mean height SDSs in girls and d) Mean BMI percentile in girls. SDS, standard deviation score; n.s., not significant; BMI, body mass index; *P*-value for the trend as determined by the Jonckheere–Terpstra test. a, *P* = 0.0499; b, *P* = 0.0003; c, *P* = 0.014; d, *P*<0.0001, e, *P* = 0.0055.

children could increase their risk of cardiovascular disease in the future, and obesity in patients with T1DM increases their risk of developing metabolic syndrome and microvascular complications [25, 26]; therefore, paediatricians need to pay attention to obesity risk in patients receiving insulin therapy.

The spread of analogue insulins may influence improved glycaemic control, reducing severe hypoglycaemia and improving anthropometric indices. However, in 2008 and 2013, almost all patients used analogue insulins, so the effects of these insulins could not be evaluated individually. Analogue insulin therapy was introduced in Japan in 2000, and increased patients' dietary options [27]. The JSGIT intensively introduced basal bolus insulin therapy for patients with T1DM in 1993 [10]. In Japan, rapid-acting insulin analogues were introduced in 2000, and long-acting insulin analogues were introduced in 2003. Furthermore, the pharmacokinetics of insulin analogues were suitable for basal bolus therapy in childhood patients. Rapid-acting insulin analogue (e.g., lispro) blood concentrations rise sharply and drop precipitously [28] without increasing the risk of severe hypoglycaemic events [29]. The long-acting insulin analogues (e.g., glargine and detemir) enable diabetic patients to maintain peak-less basal insulin levels with fewer hypoglycaemic episodes [30, 31]. Combining glargine and detemir also enables patients to receive insulin while eating snacks. Although we did not assess patients' diets in our study, the increase that we observed in the use of bolus insulin at tea-time might suggest that a subset of the patients had excess caloric intake. In this study, although

there was no information regarding the time of onset of severe hypoglycaemia, previous studies stated that hypoglycaemia in children occurred mostly as nocturnal hypoglycaemia, and that this was more common at a younger age [32, 33]. In addition, the incidence of severe hypoglycaemia has been reduced by continuous subcutaneous insulin infusion (CSII) therapy [34]. As in our study, Yamamoto et al reported that the use of CSII increased in all age groups, especially in the 0–5-year age group [35]. This might be due to a decrease in nocturnal hypoglycaemia because of the increased use of CSII in younger children and the shift from conventional insulin to insulin analogues. It is possible that positive attitudes towards treatment, including insulin therapy, progressed because of a reduced fear of hypoglycaemia.

Anthropometric indices between T1DM with or without using bolus insulin at tea-time did not differ significantly. This lack of difference might be because data for bolus insulin at tea-time indicated only whether this occurred; no data were available for consistency of use and frequency during the day and dose.

There are limitations to this study. First, we were unable to track the outcomes of individual children longitudinally; however, we were able to identify secular trends in height SDSs, BMI percentile and the prevalence of obesity in the sample population. Furthermore, determining BMI percentiles enabled us to evaluate overall secular trends in addition to age-specific changes. Because T1DM is rare in Japan, it was difficult to compare increasing or decreasing trends in obesity with healthy children by age group. Second, we analysed data collected only during the summer; however, this approach enabled us to eliminate seasonal influences on the outcomes [36–38]. Third, we were unable to assess the influence of puberty on patient outcomes. The onset of puberty was not assessed because we had no data to evaluate. However, height SDSs and BMI percentile indices were based on data for Japanese children with no information on the onset of puberty, and the indices included variations in the onset of puberty within the group. Therefore, these indices could be used without considering the evaluation of individual adolescence levels. Fourth, the possibility of selection bias cannot be ruled out. The number of participants increased, as did the number of institutes, over the course of the study period. Additionally, age and the duration of diabetes in the participants differed between the four cohorts. This may have contributed to the difference in age and duration of diabetes indicated found in participants in 1995 who were older than 6 years of age, to avoid including patients with monogenic diabetes [10]. As the original institutes involved in the 1995 cohort were the core centres for paediatric diabetes, it might be undeniable that participants in those institutes were more difficult cases.

Our results suggest that the increased use of insulin analogues has reduced the risk of severe hypoglycaemic events and improved glycaemic control. However, the use of these insulins may also have resulted in excess food intake. Patients with T1DM, as well as type 2 diabetes, need to be educated about appropriate caloric intake, nutritional composition and BMI.

## Conclusions

Japanese paediatric patients with T1DM exhibited increasing trends in height SDSs and BMI percentile from 1995 to 2013. These results could be due to improvements in patients' conditions because of the availability of new treatment options. While glycaemic control has improved and the incidence of severe hypoglycaemia has decreased, the proportion of obesity has increased.

## Supporting information

**S1 File. STROBE statement—checklist of items that should be included in reports of observational studies.**
(DOC)

## Acknowledgments

We would like to extend our deepest gratitude to all the patients, their families and the members of The Japanese Study Group of Insulin Therapy for Childhood and Adolescent Diabetes (JSGIT; jsgit@office-mms.jp).

The members of the JSGIT are as follows: K. Konishi of Abuyama Pediatrics Clinic; T. Hamajima of Aichi Children's Health and Medical Center; I. Takahashi (takaiku@doc.med.akita-u.ac.jp) of Akita University; T. Mukai (mukai5p@asahikawa-med.ac.jp), S. Suzuki (shige5p@asahikawa-med.ac.jp) and Y. Tanahashi (yutanaha@asahikawa-med.ac.jp) of Asahikawa Medical University; S. Miyamoto of Chiba Children's Hospital; M. Minagawa and K. Kinoshita of Chiba University; T. Tatematsu (tatematsu.ped@chubuh.johas.go.jp) of Chubu Rosai Hospital; K. Tsubouchi (k-tsubo@muh.biglobe.ne.jp) of Chuno Kosei Hospital; S. Koyama (s-koyama@dokkyomed.ac.jp), N. Shimura and T. Ayabe (ayabe-t@ncchd.go.jp) of Dokkyo Medical University; K. Kida, K. Takemoto (takemoto@m.ehime-u.ac.jp) and J. Hamada (blue1116-jetray@yahoo.co.jp) of Ehime University; H. Kohno an K. Miyako (miyako.k@fcho.jp) of Fukuoka Children's Hospital; K. Onigata (konigata@showa.gunma-u.ac.jp), T. Kowase and F. Mizoguchi of Gunma University; H. Ogawa, K. Nasuda, K. Otaka and T. Ogata (tomogata@-hama-med.ac.jp) of Hamamatsu University; K. Jinno (k-jinno@hph.pref.hiroshima.jp) of Hiroshima Prefectural Hospital; Y. Nishi (ynishi559@gmail.com) of Hiroshima Red Cross Hospital; Y. Igarashi of Igarashi Children's Clinic; T. Hirano of Ibaraki Children's Hospital; Z. Kizaki of the Japanese Red Cross Kyoto Daiichi Hospital; A. Nishii of JR Sendai Hospital; M. Okajima and Y. Kasahara of Kanazawa University; K. Shimura (k_4646_0917@hotmail.com) of Kawasaki Municipal Hospital; K. Kitta (kitta_k@hotmail.com), Y. Yokota (kitta_k@hotmail.com) and S. Ohtsu (ohtsukitasato@yahoo.co.jp) of Kitasato University; O. Nukada of Kobe City West Municipal Hospital; S. Konda (skykonds@camel.plala.or.jp) of Konda Children's Clinic; A. Koike (ronchan@di.mbn.or.jp) of Koike Child Clinic; T. Okada (tiger.okada@nifty.ne.jp) of Kumamoto Hatsuiku Clinic; S. Nishiyama, T. Okada, N. Yano (nozaki-39@fc.kuh.kumamoto-u.ac.jp) and K. Nakamura (nakamura@kumamoto-u.ac.jp) of Kumamoto University; T. Taketani of Kurobe Municipal Hospital; H. Nakajima (hisakazu69@yahoo.co.jp), I. Ito, A. Kinugasa and K. Kosaka of Kyoto Prefectural University of Medicine; K. Ishii (k-kanako@momo.so-net.ne.jp) and K. Ihara (k-jinno@hph.pref.hiroshima.jp) of Kyushu University; Y. Kawada (kawada@med.uoeh-u.ac.jp) of Kyushu Rosai Hospital; S. Matsuo of Matsuo Child Clinic; M. Takesue of Musashino Red Cross Hospital; Y. Amano of Nagano Red Cross Hospital; H. Mizuno (hamizuno@med.nagoya-cu.ac.jp) of Nagoya City University Hospital; R. Horikawa (horikawa-r@ncchd.go.jp) and H. Tanaka of the National Center for Child Health and Development; S. Fujitsuka and A. Motegi of the National Defence Medical Collage; S. Miyagawa (miyagawas@kure-nh.go.jp) of the National Hospital Organization Kure Medical Center; M. Nakayam (acn33450@par.odn.ne.jp) and T. Okada of Morinoki Hospital; S. Kan (sugas@mie-m.hosp.go.jp), H. Masuda and T. Fujisawa of National Mie Hospital; Y. Abe (y.abe.hosp.niigata@gmail.com) of Niigata City General Hospital; T. Mori of the National Hospital Organization Shinshu Ueda Medical Center; Y. Ohki and H. Tajima (s7047@nms.ac.jp) of Nippon Medical School; M. Inoue (masanamimoe@yahoo.co.jp) of Okayama Red Cross General Hospital; K. Hasegawa (haseyan@md.okayama-u.ac.jp) of Okayama University; T. Mochizuki (m4601256@yahoo.co.jp) of Osaka City General Medical Center; G. Isshiki and T. Kawamura (kawam@med.osaka-cu.ac.jp) of Osaka City University Graduate School of Medicine; R. Takaya (ped013@poh.osaka-med.ac.jp) of Osaka Medical College; H. Matsuura (matsuura@shinshu-u.ac.jp) and Y. Hara (youhara@shinshu-u.ac.jp) of Shinshu University; Y. Yamamoto (y-yuki@med.uoeh-u.ac.jp) of University of Occupational and Environmental Health; K. Aizu (aizu.katsuya@scmc.pref.saitama.jp) and H. Mochizuki (mochizuki.

hiroshi@pref.saitama.lg.jp) of Saitama Children's Medical Center; A. Ohtake (akira_oh@saitama-med.ac.jp) of Saitama Medial University; N. Fukushima of Sapporo City General Hospital; M. Fujimoto of St. Marianna University School of Medicine; K. Minamitani (kminami@med.teikyo-u.ac.jp) of Teikyo University Chiba Medical Center; S. Teno of Teno Clinic; J. Sugano, E. Ogawa and I. Fujiwara (ifujiwara-endo@umin.ac.jp) and J. Kanno (junko-kan@ya2.so-net.ne.jp) of Tohoku University Hospital; O. Shinohra of Tokai University; M. Tokuda of Tokuda Children's Clinic; T. Miyoshi and Y. Kotani (aayumiko@clin.med.tokushima-u.ac.jp) of Tokushima University; I. Yokota (yokotai@shikoku-med.jp) of Shikoku Medical Center for Children and Adults; K. Muroya (kmuroya@kcmc.jp) M. Adachi and K. Tachibana of Kanagawa Children's Medical Center; T. Takahashi (oc912@miracle.ocn.ne.jp) of Takahashi Clinic; G. Sasaki (sasakig@tdc.ac.jp) of Tokyo Dental College Ichikawa General Hospital; U. Sato (unishizawa-tky@umin.ac.jp) of Tokyo Hitachi Hospital; Y. Hasegawa of Tokyo Kiyose Metropolitan Children's Hospital; T. Isojima (isojimat@gmail.com), Y. Miki and S. Kitanaka (sachi-tky@umin.ac.jp) of Tokyo University School of Medicine; K. Sasaki (drsasaki@bb.mbn.or.jp) of Tokyo Women's Medical University School of Medicine Yachiyo Medical Center; Y. Uchigata (uchigata@dmc.twmu.ac.jp) and J. Miura (jmiura.dmc@twmu.ac.jp) of Tokyo Women's Medical University School of Medicine Diabetes Center; S. Kanematsu and E. Tachikawa (tckw0604@yahoo.co.jp) of Tokyo Women's Medical University School of Medicine; T. Hotsubo of Tonan Hospital; K. Hanaki (hanaki-k@umin.ac.jp) of Tottori Prefectural Kousei Hospital; M. Fujimoto and S. Kanzaki (smkanzak@grape.med.tottori-u.ac.jp) of Tottori University Faculty of Medicine; M. Uchiyama of University Graduate School of Medical and Dental Sciences; K. Kobayashi (kisho@kobayashikids.com) of the University of Yamanashi; R. Kizu (kzr_0128y@yahoo.co.jp) of Yokosuka Kyosai Hospital; K. Shiga (kshiga8@yokohama-cu.ac.jp) and N. Kikuchi (kikuchi0505@icloud.com) of the Yokohama City University Hospital; T. Urakami (urakami.tatsuhiko@nihon-u.ac.jp) and J. Suzuki (suzuki.junichi@nihon-u.ac.jp) of Nihon University; A. Endo (aendoh@hospital.iwata.shizuoka.jp) of Iwata Municipal Hospital; T. Hasegawa (haseyan@md.okayama-u.ac.jp) of Okayama University; T. Hori (tomohirohori56120@yahoo.co.jp) of the University of Gifu; Y. Ogawa (yohei_oga@yahoo.co.jp) and K. Nagasaki (nagasaki@med.niigata-u.ac.jp) of Niigata University; Y. Yasuda (yasuda.yuki@twmu.ac.jp) of Tokyo Women's Medical University School of Medicine, Center of East; S. Yatsuga (bluemif@gmail.com) and Y. Okamatsu (okamatsu30@gmail.com) of Iizuka Hospital; K. Sato (satok@major.ocn.ne.jp) of Sapporo Factory Kids Clinic; S. Kadoya (ped03@nishi-hp.jp) of Nishinomiya Municipal Central Hospital.

We thank Emily Crow, PhD, and Jane Charbonneau, DVM, from Edanz Group (https://en-author-services.edanzgroup.com/ac) for editing a draft of this manuscript.

## Author Contributions

**Conceptualization:** Mie Mochizuki, Toru Kikuchi, Ikuma Musha, Shigetaka Sugihara, Shin Amemiya.

**Data curation:** Mie Mochizuki, Yoshiya Ito, Hiroshi Yokomichi, Toru Kikuchi, Shun Soneda, Ikuma Musha, Makoto Anzou, Koji Kobayashi.

**Formal analysis:** Mie Mochizuki, Hiroshi Yokomichi.

**Funding acquisition:** Mie Mochizuki, Hiroshi Yokomichi, Toru Kikuchi.

**Investigation:** Mie Mochizuki, Toru Kikuchi, Shun Soneda, Ikuma Musha, Makoto Anzou, Koji Kobayashi, Shigetaka Sugihara, Nozomu Sasaki, Nobuo Matsuura, Shin Amemiya.

**Methodology:** Mie Mochizuki, Yoshiya Ito, Hiroshi Yokomichi, Toru Kikuchi, Makoto Anzou.

**Project administration:** Mie Mochizuki, Yoshiya Ito, Toru Kikuchi, Shigetaka Sugihara, Nozomu Sasaki, Nobuo Matsuura, Shin Amemiya.

**Resources:** Mie Mochizuki, Toru Kikuchi, Shigetaka Sugihara, Nozomu Sasaki, Nobuo Matsuura, Shin Amemiya.

**Software:** Mie Mochizuki.

**Supervision:** Mie Mochizuki, Yoshiya Ito, Hiroshi Yokomichi, Toru Kikuchi, Shigetaka Sugihara, Nozomu Sasaki, Nobuo Matsuura, Shin Amemiya.

**Validation:** Mie Mochizuki, Hiroshi Yokomichi, Toru Kikuchi, Ikuma Musha, Makoto Anzou, Shin Amemiya.

**Visualization:** Mie Mochizuki.

**Writing – original draft:** Mie Mochizuki.

**Writing – review & editing:** Mie Mochizuki, Hiroshi Yokomichi, Toru Kikuchi, Shun Soneda, Ikuma Musha, Makoto Anzou, Koji Kobayashi, Kumihiro Matsuo, Shigetaka Sugihara, Nozomu Sasaki, Nobuo Matsuura, Shin Amemiya.

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
