## [Decision Letter · Decision Letter 0]

7 Jul 2020

PONE-D-20-12969

Increasing secular trends in height and obesity in children with type 1 diabetes: JSGIT Cohort

PLOS ONE

Dear Dr. Mocizuki,

Thank you for submitting your manuscript to PLOS ONE. After careful consideration, we feel that it has merit but does not fully meet PLOS ONE’s publication criteria as it currently stands. Therefore, we invite you to submit a revised version of the manuscript that addresses the points raised during the review process.

We look forward to receiving your revised manuscript.

Kind regards,

Yongfu Yu, Ph.D

Academic Editor

PLOS ONE

Additional Editor Comments:

The audience would benefit from a description of JSGIT cohorts.Any rationale for the use of the cutoff points 58 and 75 mmmol/mol, from the ISPAD guideline?Please think about using graph to visualize the results.Please rewrite the first paragraph of discussion, the audience does not need to read the table to know the key message of this study

Journal Requirements:

3. One of the noted authors is a group, JSGIT. In addition to naming the author group and listing the individual authors and affiliations within this group in the acknowledgments section of your manuscript, please also indicate clearly a lead author for this group along with a contact email address.

Reviewers' comments:

Reviewer's Responses to Questions

**Comments to the Author**

1. Is the manuscript technically sound, and do the data support the conclusions?

Reviewer #1: Yes

Reviewer #2: Yes

Reviewer #3: Partly

2. Has the statistical analysis been performed appropriately and rigorously? 

Reviewer #1: Yes

Reviewer #2: Yes

Reviewer #3: Yes

3. Have the authors made all data underlying the findings in their manuscript fully available?

Reviewer #1: No

Reviewer #2: No

Reviewer #3: Yes

4. Is the manuscript presented in an intelligible fashion and written in standard English?

Reviewer #1: No

Reviewer #2: Yes

Reviewer #3: Yes

5. Review Comments to the Author

Reviewer #1: Thank you for giving me an opportunity to review this article. The authors investigated trends in anthropometric measurements among Japanese pediatric patients with Type 1 diabetes. This descriptive study is important and helpful for clinicians and public health professionals to understand the current trends and situations for Japanese patients with the understudied endocrine disorder. However, the authors might want to have more careful interpretation and thorough discussion of their findings.

Major comments

1. Line 55. “Based on this improvement, Japanese guidelines have been adjusted..”. Could the authors cite the most recent Japanese guideline? Was it published after the cited articles published in 2017 (Mochizuki et al. and Yokomichi et al.)? If not, the authors might want to cite the evidence “based on” which the Japanese guideline was adjusted to HbA1c <7.5.

2. Line 111. What was the reason of the increase in the number of T1DM patients in this study (513�685�734�978)? Was there selection bias? For example, would it be possible that patients enrolled in earlier years were more likely to be severe (or have a high percentage of hypoglycemia) compared with patients enrolled in later years? Please clarify the underlying reason for these trends in the discussion.

3. Table 1. The reduction of hypoglycemia events from 2000 to 2008 was surprisingly high. What was the underlying reason? Why do the authors think severe hypoglycemic events decreased although insulin use increased? This might be discussed in the authors’ previous paper, but further explanation would be needed in the discussion for readers of this paper.

4. Table 1. Likewise, what is the underlying reason for the substantial increase in the proportion of patients using insulin analog and patients using bolus insulin at tea-time?

It looks that the cohort in 2000 is pretty different from the cohort in 2008 and 2013.

5. Table 2. I’m not sure I agree with the authors’ opinion about the trends in SDS and BMI. Although they are ‘statistically significant’, the row numbers do not necessarily show the monotonic trends. For example, SDS fluctuate among patients with HbA1c 7.5-8.9% (-0.12�-0.17�+0.08�-0.08) and HbA1c >9% (-0.35�-0.18�-0.32�-0.13). BMI among patients with HbA1c >9% increased in 2000 but then plateaued since 2000. The authors may need to interpret the results based on the real trends rather than `statistical trend analysis’ which could be significant if there is some discrepancy across the year. A graphical presentation might be an alternative option to show the exact trends. Please revise the discussion on these findings as well.

6. Line 169. I agree with this statement (the rarity of T1DM did not allow them to investigate the trends by age group). However, a comparison of overall trends with healthy children would be feasible and informative if the data is available. Please consider adding them. Even if the data itself is not available, discussion comparing the published trends in healthy children would be important.

7. Line 174. “The increased use of insulin analog has reduced the risk of severe hypoglycemic events and improved glycemic control.” Please rephrase this sentence. This study does not tell anything about causality (i.e. reduce or improve something), but only tells the descriptive trends. Moreover, again, why do the authors think they observed the increased use of insulin but the decreased severe hypoglycemic events? This might be counterintuitive for some readers, and needs to be adequately discussed with some evidence.

Minor comments

1. Line 54. “Recent studies have reported a decreased in the mean HbA1c of patients with T1DM from 7.7% to 8.4%”. Is this typo? The number increased from 7.7 to 8.4 although they mentioned “decrease”.

2. Line 72. As most of the readers are not familiar with JSGIT, please explain details of this cohort (is this national survey representing Japanese national population?), at least in Appendix.

3. “tea” bolus insulin may not be the appropriate word. I would suggest using the phrase “using bolus insulin at tea-time” throughout the manuscript.

4. Line 139. Citing Tables in the discussion is not common.

5. Line 172. Is there evidence about seasonal influences? If so, please cite them.

6. Line 172. Why did the author mention only puberty? Other outcomes?

Reviewer #2: The authors are presenting trends in anthropometric indices of the four cohorts of Japanese children with type 1 diabetes mellitus (T1DM) over the period of two decades. They successfully demonstrated improved glycemic control and height as the therapeutic modalities had been changing. At the same time, however, BMI percentiles and rate of obesity were shown to be increased especially in the subgroups of poor glycemic control. Although these findings may have been expected, this is probably the first to show in the serial cohorts conducted by the same study group. To achieve both better glycemic control and healthy constitution in growing children with T1DM, these findings will give clinicians important messages.

1. P1 L5: Nobuo Matsuura’s affiliation may be wrong.

2. P3 L47: “increasing insulin doses” do not seem to be correct, as the total insulin doses are stable as shown in Table 1. The authors may have intended to mean “proportion of patients using tea bolus insulin”.

3. P4 L51-57: This paragraph is obscure. The authors should indicate clearly whether they are discussing about only children (or patients of all ages) and Japanese (or worldly) situation.

4. P4 L55: A reference(s) for “the Japanese guidelines” is needed.

5. P5 L67: Should “a cohort” read “the Japanese cohorts”?

6. P5 L75: Please eliminate {‘} from {Patients’}.

7. P5 L82: The symbol < is lacking before “7.5%”.

8. P5 L83: The symbol > before “9.0%” is more correctly ≥.

9. P5 L92: Use of a single national reference is understandable to clarify the trend of anthropometric indices of specific groups over a certain period. At the same time, however, comparison with the trend seen in the general population is also important. Otherwise, the interpretation, i.e. the conclusion of this paper, may become misleading.

10. P7 L111-118 and Table 1: The age distribution of each cohort should be described as the indices of overweight may be influenced by age.

11. P8 Table 1: As explained above (9.), trends in mean BMI percentile and mean BMI SDS in general population ae also important. The timing of puberty may have been still changing over the period between 1995 and 2013 cohorts. Accordingly, in general population, the BMI-related indices at the particular typical age of these cohorts may be also have a similar trend. Please discuss about this possibility.

12. P9 L12-131: The authors are discussing about the trend in anthropometric indices by glycemic control groups. The readers are also interested in whether a risk of obesity is increased along with puberty, because the demand of insulin obviously increases at the time. Please try to add a table and discussion regarding the trend in anthropometric indices by age groups.

13. P10 L151-152: The assumption may be correct but it is not shown that the dose of insulin in high-risk glycemic control groups is more than in the better control group in this study.

14. P10 L155: “The increased use of insulin analog” is obscure. Are the authors discussing about dose or frequency of which kind of insulin analog?

15. P11 L163-165: If the authors would like to discuss about the contribution of tea bolus insulin to the tendency of obesity, they should analyze BMI-related indices in patients with or without tea bolus insulin in the cohorts after 2000.

Reviewer #3: This is very impressive and important study to explore increasing secular trends in height and obesity in children with type 1 diabetes. The main strength of this study is to analyze serum haemoglobin A1c (HbA1c) levels, the incidence of severe hypoglycemic events, the types and doses of insulin, height standard deviation scores (SDS), body mass index (BMI) percentiles compared with healthy Japanese children and obesity prevalence over time. The discussion is thorough and well written. However, we suggested further discussion of the definition of the study and detailed revision of some mistakes to refine the study. The following suggestions are provided to enhance the importance of this work.

1、 We consider the study a cross sectional study rather than a cohort one. It may be more like four cross-sectional surveys.

2、 In this study, demographic features like gender and age of T1DM patients were not stated, however in certain cases the age when you get DM might be a key factor that influences BMI, obesity as well as the progression of DM. In addition, the percentages of boys recruited in 4 designated year were given, but a chi square test was suggested to speculate the statistical significance on gender proportions.

3、 Why total daily insulin dose was going down in this study while insulin analogue was becoming more popular and people tended to have excess calorie intake as what was stated in the discussion.

4、 As was mentioned that ‘SDS values were calculated based on data from healthy Japanese children record the National Survey in 2000’, reference might be needed to support the calculation.

5、 In line 54,‘a decrease in the mean haemoglobin A1c (HbA1c) level of patients with T1DM from 7.7% to 8.2%.(1, 3, 4)’should be revised as ‘a decrease in the mean haemoglobin A1c (HbA1c) level of patients with T1DM from 8.2to7.7%.’.

6、 In Results, ‘685 from 2008‘ should be revised as ‘685 from 2000’.

7、 In Conclusions: Clinicians should be aware of increased BMI in these patients and the associated risk of developing cardiovascular disease in the future. We suggested not to mention cardiovascular disease because no variables in the study were associated with cardiovascular disease.

8、 How did BMI, over weight and obesity change in the whole population from 1995 to 2013 in Japan? The factors, including confounding ones, required detailed discussion to demonstrate the association with increasing BMI or prevalence of obesity.

6. PLOS authors have the option to publish the peer review history of their article (what does this mean?). If published, this will include your full peer review and any attached files.

Reviewer #1: No

Reviewer #2: No

Reviewer #3: **Yes: **Fujie Shen

---

## [Author Response · Author response to Decision Letter 0]

15 Sep 2020

Responses to the Editor and Reviewers 

We thank the reviewers for your valuable comments and questions and for the opportunity to improve our paper. 

As requested, we prepared a revised version of our manuscript and hope that we have addressed the concerns of reviewers #1−3 and editor. Our point-by-point responses to the reviewers’ comments follow, in this letter. The reviewers’ comments are numbered according to each reviewer’s number, and our responses are designated according to the line numbers in the revised manuscript. 

We sincerely thank the reviewers for all their helpful comments.

Responses to the comments of reviewer #1

Major comments

1. Line 55. “Based on this improvement, Japanese guidelines have been adjusted.”. Could the authors cite the most recent Japanese guideline? Was it published after the cited articles published in 2017 (Mochizuki et al. and Yokomichi et al.)? If not, the authors might want to cite the evidence “based on” which the Japanese guideline was adjusted to HbA1c <7.5.

Response: We apologize for the confusion. We revised the indicated sentences as follows: “The Japanese Society for Pediatric Endocrinology (JSPE) translated these guidelines into Japanese in 2008. (6) The JSPE and the Japan Diabetes Society (JDS) recommended an HbA1c levels of < 7.5% for optimal glycaemic control since 2011, consistent with the ISPAD recommendations.(7) The Japanese Study Group of Insulin Therapy for Childhood and Adolescent Diabetes (JSGIT) uses this target in Japanese children and adolescents with T1DM. “ (Line 59) 

2. Line 111. What was the reason of the increase in the number of T1DM patients in this study (513�685�734�978)? Was there selection bias? For example, would it be possible that patients enrolled in earlier years were more likely to be severe (or have a high percentage of hypoglycemia) compared with patients enrolled in later years? Please clarify the underlying reason for these trends in the discussion.

Response: We appreciate your insightful comment regarding clarifying the description of the registered patients. We clarified the numbers of institutions, and the numbers and characteristics of the patients who were registered in the cohorts. We also created a new table and added the following sentences to the revised manuscript:

“The JSGIT is the only multi-institutional joint research group in Japan, and the group was established in 1995 for patients with childhood-onset type 1 diabetes.(10) The first cohort study began in 1995 and was designed to improve the treatment and glycaemic control of diabetic children in Japan. The number of participating institutions and registered patients is increasing cohort by cohort. The current study was a cross-sectional study performed using data from the entry year for the four cohort studies. We analysed patients’ baseline data from cohorts recruited in 1995, 2000, 2008 and 2013, and we excluded data from patients who had T1DM for less than 6 months. Patients were diagnosed with T1DM by their paediatricians according to the criteria established by the JDS and the American Diabetes Association.(11, 12) Participants’ clinical characteristic are shown in Table 1. Clinical characteristic was shown in Table 1.” (Line81)

In the discussion, we added: “Fourth, the possibility of selection bias cannot be ruled out. The number of participants increased, as did the number of institutes, over the course of the study period. Additionally, age and the duration of diabetes in the participants differed between the four cohorts. This may have contributed to the difference in age and duration of diabetes indicated found in participants in 1995 who were older than 6 years of age, to avoid including patients with monogenic diabetes.(10) As the original institutes involved in the 1995 cohort were the core centres for paediatric diabetes, it might be undeniable that participants in those institutes were more difficult cases.” (Line 282)

Table 1. Participants’ characteristics. 

3. Table 1. The reduction of hypoglycemia events from 2000 to 2008 was surprisingly high. What was the underlying reason? Why do the authors think severe hypoglycemic events decreased although insulin use increased? This might be discussed in the authors’ previous paper, but further explanation would be needed in the discussion for readers of this paper.

Response: We appreciate your advice to improve the discussion. We made the following changes to the text: “In this study, although there was no information regarding the time of onset of severe hypoglycaemia, previous studies stated that hypoglycaemia in children occurred mostly as nocturnal hypoglycaemia, and that this was more common at a younger age.(33, 34) In addition, the incidence of severe hypoglycaemia has been reduced by continuous subcutaneous insulin infusion (CSII) therapy. (35) As in our study, Yamamoto et al reported that the use of CSII increased in all age groups, especially in the 0–5-year age group. (36) This might be due to a decrease in nocturnal hypoglycaemia because of the increased use of CSII in younger children and the shift from conventional insulin to insulin analogues. It is possible that positive attitudes towards treatment, including insulin therapy, progressed because of a reduced fear of hypoglycaemia.” (Line 257)

4. Table 1. Likewise, what is the underlying reason for the substantial increase in the proportion of patients using insulin analog and patients using bolus insulin at tea-time?　It looks that the cohort in 2000 is pretty different from the cohort in 2008 and 2013.

Response: We appreciate your advice to improve the discussion. We shortened the discussion as follows: “The spread of analogue insulins may influence improved glycaemic control, reducing severe hypoglycaemia and improving anthropometric indices. However, in 2008 and 2013, almost all patients used analogue insulins, so the effects of these insulins could not be evaluated individually. Analogue insulin therapy was introduced in Japan in 2000, and increased patients’ dietary options.(28) The JSGIT intensively introduced basal bolus insulin therapy for patients with T1DM in 1993.(10) In Japan, rapid-acting insulin analogues were introduced in 2000, and long-acting insulin analogues were introduced in 2003. Furthermore, the pharmacokinetics of insulin analogues were suitable for basal bolus therapy in childhood patients. Rapid-acting insulin analogue (e.g., lispro) blood concentrations rise sharply and drop precipitously(29) without increasing the risk of severe hypoglycaemic events.(30) The long-acting insulin analogues (e.g., glargine and detemir) enable diabetic patients to maintain peak-less basal insulin levels with fewer hypoglycaemic episodes.(31, 32) Combining glargine and detemir also enables patients to receive insulin while eating snacks. Although we did not assess patients’ diets in our study, the increase that we observed in the use of bolus insulin at tea-time suggests that a subset of the patients had excess caloric intake.” (Line 244)

5. Table 2. I’m not sure I agree with the authors’ opinion about the trends in SDS and BMI. Although they are ‘statistically significant’, the row numbers do not necessarily show the monotonic trends. For example, SDS fluctuate among patients with HbA1c 7.5-8.9% (-0.12�-0.17�+0.08�-0.08) and HbA1c >9% (-0.35�-0.18�-0.32�-0.13). BMI among patients with HbA1c >9% increased in 2000 but then plateaued since 2000. The authors may need to interpret the results based on the real trends rather than `statistical trend analysis’ which could be significant if there is some discrepancy across the year. A graphical presentation might be an alternative option to show the exact trends. Please revise the discussion on these findings as well.

Response: We appreciate your insightful comment on clarifying these points in our manuscript. Accordingly, we added Figure 2, which shows the differences in height SDSs, BMI percentile and the proportion of obesity according to glycaemic control. The difference in the proportion of obesity from 2008 to 2013 might be related to a similar trend in the general population.

We revised the text as follows:

“The proportion of overweight children (weight > 20% over the standard weight adjusted for sex and height) in the Japanese general population increased from 1995 to 2003, but did not change from 2003 to 2013.(23) The proportion of overweight and obese children with T1DM in this study’s cohorts increased. The difference in the proportion of obesity from 2008 to 2013 might be associated with having diabetes.” (Line 220)

Fig. 2 Secular changes in total daily insulin dose, proportion of obesity, height SDSs and BMI percentile according to the level of glycaemic control in Japanese children with type 1 diabetes mellitus.

6. Line 169. I agree with this statement (the rarity of T1DM did not allow them to investigate the trends by age group). However, a comparison of overall trends with healthy children would be feasible and informative if the data is available. Please consider adding them. Even if the data itself is not available, discussion comparing the published trends in healthy children would be important.

Response: Thank you for this important suggestion. We cited a new reference and added the following paragraphs: “The proportion of overweight children (weight > 20% over the standard weight adjusted for sex and height) in the Japanese general population increased from 1995 to 2003, but did not change from 2003 to 2013.(23) The proportion of overweight and obese children with T1DM in this study’s cohorts increased. The difference in the proportion of obesity from 2008 to 2013 might be associated with having diabetes.” (Line 220)

7. Line 174. “The increased use of insulin analog has reduced the risk of severe hypoglycemic events and improved glycemic control.” Please rephrase this sentence. This study does not tell anything about causality (i.e. reduce or improve something), but only tells the descriptive trends. Moreover, again, why do the authors think they observed the increased use of insulin but the decreased severe hypoglycemic events? This might be counterintuitive for some readers, and needs to be adequately discussed with some evidence. 

Response: We appreciate your advice regarding improving the discussion. We agree that this study showed trends and not causal relationships. In 2008 and 2013, almost all patients used analogue insulins, so the effects of these insulins could not be evaluated individually. We shortened the Discussion section as follows: “The spread of analogue insulins may influence improved glycaemic control, reducing severe hypoglycaemia and improving anthropometric indices. However, in 2008 and 2013, almost all patients used analogue insulins, so the effects of these insulins could not be evaluated individually. Analogue insulin therapy was introduced in Japan in 2000, and increased patients’ dietary options.(28) The JSGIT intensively introduced basal bolus insulin therapy for patients with T1DM in 1993.(10) In Japan, rapid-acting insulin analogues were introduced in 2000, and long-acting insulin analogues were introduced in 2003. Furthermore, the pharmacokinetics of insulin analogues were suitable for basal bolus therapy in childhood patients. Rapid-acting insulin analogue (e.g., lispro) blood concentrations rise sharply and drop precipitously(29) without increasing the risk of severe hypoglycaemic events.(30) The long-acting insulin analogues (e.g., glargine and detemir) enable diabetic patients to maintain peak-less basal insulin levels with fewer hypoglycaemic episodes.(31, 32) Combining glargine and detemir also enables patients to receive insulin while eating snacks. Although we did not assess patients’ diets in our study, the increase that we observed in the use of bolus insulin at tea-time suggests that a subset of the patients had excess caloric intake.” (Line 244)

Minor comments

1. Line 54. “Recent studies have reported a decreased in the mean HbA1c of patients with T1DM from 7.7% to 8.4%”. Is this typo? The number increased from 7.7 to 8.4 although they mentioned “decrease”.

Response: We apologize for the confusion. We rewrote the indicated sentence as follows: “Although glycaemic control varies among countries, age groups and patients,(2) recent studies have reported a decreased mean haemoglobin A1c (HbA1c) level in Japanese patients with T1DM of between 7.7% and 8.2%.(1, 3, 4)” (Line 54)

2. Line 72. As most of the readers are not familiar with JSGIT, please explain details of this cohort (is this national survey representing Japanese national population?), at least in Appendix.{Yokoya, 2014 #959}

Response: Thank you for this helpful suggestion, which the journal’s Academic Editor also made. We added information about the JSGIT and a new reference, as follows: “The JSGIT is the only multi-institutional joint research group in Japan, and the group was established in 1995 for patients with childhood-onset type 1 diabetes.(10) The first cohort study began in 1995 and was designed to improve the treatment and glycaemic control of diabetic children in Japan. The number of participating institutions and registered patients is increasing cohort by cohort.” (Line 81)

Ref. 10: Matsuura N, Yokota Y, Kazahari K, Sasaki N, Amemiya S, Ito Y, et al. The Japanese Study Group of Insulin Therapy for Childhood and Adolescent Diabetes (JSGIT): initial aims and impact of the family history of type 1 diabetes mellitus in Japanese children. Pediatric diabetes. 2001;2(4):160-9.

3. “tea” bolus insulin may not be the appropriate word. I would suggest using the phrase “using bolus insulin at tea-time” throughout the manuscript.

Response: We appreciate your advice to clarify the meaning of additional/tea bolus insulin. We changed “tea bolus insulin” to “using bolus insulin at tea-time” in Table 2. We also added the following: “the proportion of patients using bolus insulin at tea-time” (Line Line123, 152 and 206)

4. Line 139. Citing Tables in the discussion is not common.

Response: Thank you very much for your comment. We deleted “Table 1 and Table 2” in the first paragraph of the discussion, in accordance with your suggestion. (Line 203)

5. Line 172. Is there evidence about seasonal influences? If so, please cite them.

Response: We appreciate your recommendation to cite appropriate evidence in the discussion. We cited three references indicating seasonal variation in weight gain and glycaemic control. (Line 277)

Ref. 37. Mianowska B, Fendler W, Szadkowska A, Baranowska A, Grzelak-Agaciak E, Sadon J, et al. HbA(1c) levels in schoolchildren with type 1 diabetes are seasonally variable and dependent on weather conditions. Diabetologia. 2011;54(4):749-56.

Ref. 38. Marshall ELJJoP. A review of American research on seasonal variation in stature and body weight. 1937;10:819-31.

Ref. 39. Kobayashi M, Kobayashi MJE, Biology H. The relationship between obesity and seasonal variation in body weight among elementary school children in Tokyo. 2006;4(2):253-61.

6. Line 172. Why did the author mention only puberty? Other outcomes?

Response: We appreciate this comment. Puberty was not an outcome in this study. To clarify, we added the following: “The onset of puberty was not assessed because we had no data to evaluate. However, height SDSs and BMI percentile indices were based on data for Japanese children with no information on the onset of puberty, and the indices included variations in the onset of puberty within the group. Therefore, these indices could be used without considering the evaluation of individual adolescence levels.” (Line 278)

Responses to the comments of reviewer #2

1. P1 L5: Nobuo Matsuura’s affiliation may be wrong.

Response: As you commented, Nobuo Matsuura was a professor at Kitasato University and Seitoku University. He retired from these institutions in 2019 and is now affiliated with Bibai City Hospital in Hokkaido. We made the appropriate changes, in the revised manuscript.

2. P3 L47: “increasing insulin doses” do not seem to be correct, as the total insulin doses are stable as shown in Table 1. The authors may have intended to mean “proportion of patients using tea bolus insulin”.

Response: We noticed that an entry in Table 1 was incorrect, in the original manuscript. We apologize for this error and revised a new Table 2 as follows: We changed “1.10“to “1.01” in the 1995 cohort column. (Line 152)

3. P4 L51-57: This paragraph is obscure. The authors should indicate clearly whether they are discussing about only children (or patients of all ages) and Japanese (or worldly) situation.

Response: Thank you for your recommendation to clarify. We rewrote the indicated paragraph as follows: “Although glycaemic control varies among countries, age groups and patients,(2) recent studies have reported a decreased mean haemoglobin A1c (HbA1c) level in Japanese patients with T1DM of between 7.7% and 8.2%.(1, 3, 4) The 2007 International Society for Pediatric and Adolescent Diabetes (ISPAD) consensus guidelines management objectives indicated that unless there was severe hypoglycaemia and repeated hypoglycaemia, the target HbA1c for all children under the age of 18 was < 7.5%. (5) The Japanese Society for Pediatric Endocrinology (JSPE) translated these guidelines into Japanese in 2008. (6) The JSPE and the Japan Diabetes Society (JDS) recommended an HbA1c levels of < 7.5% for optimal glycaemic control since 2011, consistent with the ISPAD recommendations.(7) The Japanese Study Group of Insulin Therapy for Childhood and Adolescent Diabetes (JSGIT) uses this target in Japanese children and adolescents with T1DM.” (Line 54)

4. P4 L55: A reference(s) for “the Japanese guidelines” is needed.

Response: Thank you for your comment. We added a reference to the guidelines for the Japan Diabetes Society and The Japanese Society for Pediatric Endocrinology, as follows: “The goal of glycemic control. Guide for treatment of childhood and adolescent diabetics. 2011;3rd revision:130-8.” as Ref.6.

Ref.6. Rewers M, Pihoker C, Donaghue K, Hanas R, Swift P, Klingensmith G. The International Society for Pediatric and Adolescent Diabetes. Clinical Practice Consensus Guidelines 2006-2008: Chapter 7 Assessment and monitoring of glycemic control. The Journal of the Japan Pediatric Society. 2008;112(9):810-20.

5. P5 L67: Should “a cohort” read “the Japanese cohorts”?

Response: Thank you for your advice. We corrected the indicated phrase, as “cohorts of Japanese paediatric patients.” (Line 75)

6. P5 L75: Please eliminate {‘} from {Patients’}.

Response: Thank you for your comment. We eliminated {‘} from {Patients’}. (Line 87)

7. P5 L82: The symbol < is lacking before “7.5%”

Response: Thank you for your comment. We added the symbol < before “7.5%”. (Line108)

8. P5 L83: The symbol > before “9.0%” is more correctly ≥,

Response: Thank you for your helpful comment. We changed the symbol > before “9.0%” to ≥. (Line 110)

9. P5 L92: Use of a single national reference is understandable to clarify the trend of anthropometric indices of specific groups over a certain period. At the same time, however, comparison with the trend seen in the general population is also important. Otherwise, the interpretation, i.e. the conclusion of this paper, may become misleading.

Response: Thank you for this important suggestion. We cited an appropriate reference and added the following: “The proportion of overweight children (weight > 20% over the standard weight adjusted for sex and height) in the Japanese general population increased from 1995 to 2003, but did not change from 2003 to 2013.(23) The proportion of overweight and obese children with T1DM in this study’s cohorts increased. The difference in the proportion of obesity from 2008 to 2013 might be associated with having diabetes.” (Line 220)

10. P7 L111-118 and Table 1: The age distribution of each cohort should be described as the indices of overweight may be influenced by age.

Response: As you commented, the indices for overweight may be influenced by age. We created a new Table 1, as follows:

Table 1. Participants’ characteristics. 

11. P8 Table 1: As explained above (9.), trends in mean BMI percentile and mean BMI SDS in general population ae also important. The timing of puberty may have been still changing over the period between 1995 and 2013 cohorts. Accordingly, in general population, the BMI-related indices at the particular typical age of these cohorts may be also have a similar trend. Please discuss about this possibility.

Response: Thank you for this important comment. The onset of puberty among Japanese children advanced from 1950 to 2010, but stabilised by 1970. We edited the indicated text, as follows: “Considering puberty, the onset of puberty among Japanese children advanced from 1950 to 2010, but stabilised by 1970. (19) Height SDSs, BMI percentile and the proportion of overweight and obese children with T1DM could not be evaluated using Japanese data for the year 2000.” (Line 225)

12. P9 L12-131: The authors are discussing about the trend in anthropometric indices by glycemic control groups. The readers are also interested in whether a risk of obesity is increased along with puberty, because the demand of insulin obviously increases at the time. Please try to add a table and discussion regarding the trend in anthropometric indices by age groups.

Response: We appreciate your insightful comment to improve our manuscript. We added new figures and additional text. We categorised patients into four age groups; a 5–6-year-old group, 10–11-year-old group, 12–13-year-old group and 15–16-year-old group.

New/revised text:

“We categorised patients into four age groups; 5–6-year-old group, 10–11-year-old group, 12–13-year-old group and 15–16-year-old group, according to sex. Pubertal stage was classified as follows: in boys, the 5–6-year-old and 10–11-year-old groups indicated the pre-pubertal stage; the 12–13-year-old group indicated the early pubertal stage and the 15–16-year-old group indicted the middle to the end of the pubertal stage. In girls, the 5–6-year-old group indicated the pre-pubertal stage; the 10–11-year-old group indicated the early pubertal stage; the 12–13-year-old group indicated the middle pubertal stage and the 15–16-year-old group indicated the end of the pubertal stage.(19)”　(Line 113)

“According to the age groups, height SDSs and BMI percentile showed significant increasing trends in the 15–16-year-old group (P=0.0003 and P=0.014 in boys and P<0.0001 and P=0.0055 in girls, respectively) (Fig. 3).” (Line 191)

Fig. 3 Secular changes in height SDSs and BMI percentile in the four age groups.

“Height SDSs and BMI percentile showed a significant increasing trend in the 15–16-year-old group for both boys and girls in the cohorts, which reflect increased height SDSs and BMI percentiles at the end of the pubertal stage.” (Line 228)

　 

13. P10 L151-152: The assumption may be correct but it is not shown that the dose of insulin in high-risk glycemic control groups is more than in the better control group in this study.

Response: Thank you for this important suggestion. We added a new figure and additional sentences in the results, as follows: “Total daily insulin dose showed an increasing trend in the order of high-risk control group, suboptimal control group and optimal control group (Fig. 2).” (Line 167)

Fig. 2 Secular changes in total daily insulin dose, proportion of obesity, height SDSs and BMI percentile according to the level of glycaemic control in Japanese children with type 1 diabetes mellitus.

14. P10 L155: “The increased use of insulin analog” is obscure. Are the authors discussing about dose or frequency of which kind of insulin analog?

Response: Thank you for your suggestion. “The increased use of insulin analog” means increased dose and frequency of insulin analogues, with a shift from NPH insulin to long-acting insulin and from regular insulin to rapid-acting insulin. To clarify, we added the following: “and a shift from NPH insulin to long-acting insulin and from regular insulin to rapid-acting insulin analogues “　 (Line 234)

15. P11 L163-165: If the authors would like to discuss about the contribution of tea bolus insulin to the tendency of obesity, they should analyze BMI-related indices in patients with or without tea bolus insulin in the cohorts after 2000.

Response: We added the following paragraphs in the results and discussion, respectively, to clarify:

“Height SDSs, BMI percentile and proportion of obesity between T1DM with or without using bolus insulin at tea-time did not differ significantly (−0.20 (1.23) vs. −0.42 (1.80), 58.1 (25.8) vs. 58.3 (24.5) and 0.0% vs. 1.8% in 2000; 0.04 (1.07) vs. 0.06 (1.09), 58.6 (25.8) vs. 56.5 (26.7) and 18.4% vs. 20.1% in 2008; 0.17 (1.07) vs. 0.17 (0.89), 59.2 (26.3) vs. 57.4 (25.5) and 28.0% vs. 26.5% in 2013, respectively).” (Line 167)

“Anthropometric indices between T1DM with or without using bolus insulin at tea-time did not differ significantly. This lack of difference might be because data for bolus insulin at tea-time indicated only whether this occurred; no data were available for consistency of use and frequency during the day and dose.” (Line 267)

Responses to the comments of reviewer #3

1. We consider the study a cross sectional study rather than a cohort one. It may be more like four cross-sectional surveys.

Response: We appreciate your insightful comment, and we have rewritten the description of the study design as “The current study was a cross-sectional study performed using data from the entry year for the four cohort studies.” (Line 85)

2. In this study, demographic features like gender and age of T1DM patients were not stated, however in certain cases the age when you get DM might be a key factor that influences BMI, obesity as well as the progression of DM. In addition, the percentages of boys recruited in 4 designated year were given, but a chi square test was suggested to speculate the statistical significance on gender proportions.

Response: We added a new table, Table 1. We also added the following: “The proportions of girls were higher in each cohort, but there were no significant differences between the cohorts. A dominant proportion of girls in a cohort is a common characteristic in Japanese patients with T1DM. (1)” (Line 188)

Table 1. Participants’ characteristics. 

3. Why total daily insulin dose was going down in this study while insulin analogue was becoming more popular and people tended to have excess calorie intake as what was stated in the discussion.

Response: We noticed that in the original manuscript, an entry in Table 1 was incorrect. We apologize for this error and corrected the value in Table 1 as follows: We changed “1.10 “to “1.01” in the 1995 cohort column. The total daily insulin dose showed an increasing trend. (Line 152)

Table 2. Secular changes in height SDSs and the prevalence of overweight and obesity in Japanese children with type 1 diabetes mellitus.

4. As was mentioned that ‘SDS values were calculated based on data from healthy Japanese children record the National Survey in 2000’, reference might be needed to support the calculation.

Response: Thank you for your suggestion. We cited an appropriate reference. 

“We defined overweight and obesity as BMI in the 85th–94th percentile and BMI ≥ 95th percentile(13), respectively, according to the 2000 JSPE data for each age and sex.(14) We also evaluated BMI standard deviation scores (SDSs) and height SDSs to compare the data from patients across all age groups.(15) SDS values for controls were calculated using data from healthy Japanese children recorded in the National Survey in 2000.(16)”

(Line 100)

Ref.16 Ministry of Education C, Sports, Science and technology. Annual Report of School Health Statistics Research 2014.

5. In line 54,‘a decrease in the mean haemoglobin A1c (HbA1c) level of patients with T1DM from 7.7% to 8.2%.(1, 3, 4)’should be revised as ‘a decrease in the mean haemoglobin A1c (HbA1c) level of patients with T1DM from 8.2to7.7%.’.

Response: We apologize for the confusion. We rewrote the indicated sentence as: “Although glycaemic control varies among countries, age groups and patients,(2) recent studies have reported a decreased mean haemoglobin A1c (HbA1c) level in Japanese patients with T1DM of between 7.7% and 8.2%.(1, 3, 4)” (Line 54)

6. In Results, ‘685 from 2008‘ should be revised as ‘685 from 2000’.

Response: Thank you for this comment. We revised as “685 from 2000”. (Line 142) 

7. In Conclusions: Clinicians should be aware of increased BMI in these patients and the associated risk of developing cardiovascular disease in the future. We suggested not to mention cardiovascular disease because no variables in the study were associated with cardiovascular disease.

Response: We deleted the sentences discussing cardiovascular disease.

8. How did BMI, over weight and obesity change in the whole population from 1995 to 2013 in Japan? The factors, including confounding ones, required detailed discussion to demonstrate the association with increasing BMI or prevalence of obesity.

Response: Thank you for your insightful comment. We added the following sentences in the discussion: “The proportion of overweight children (weight > 20% over the standard weight adjusted for sex and height) in the Japanese general population increased from 1995 to 2003, but did not change from 2003 to 2013.(23) The proportion of overweight and obese children with T1DM in this study’s cohorts increased. The difference in the proportion of obesity from 2008 to 2013 might be associated with having diabetes.” (Line 220)

Responses to the Additional Editor Comments:

1. The audience would benefit from a description of JSGIT cohorts.

Response: Thank you for the insightful comment. We added the following information about the JSGIT, and new references: “The JSGIT is the only multi-institutional joint research group in Japan, and the group was established in 1995 for patients with childhood-onset type 1 diabetes.(10) The first cohort study began in 1995 and was designed to improve the treatment and glycaemic control of diabetic children in Japan.” (Line 81)

2. Any rationale for the use of the cutoff points 58 and 75 mmmol/mol, from the ISPAD guideline?

Response; Thank you for this question. The DCCT conventional adult cohort had a mean HbA1c value of 8.9%, and both DCCT and EDIC have shown poor outcomes with this level; therefore, ≥ 9% (75 mmol/mol) was recommended as the cut-off value as high-risk. In contrast, a target of < 7.5% (58 mmol/mol) was recommended for all patients younger than 18 years in the 2006 and 2014 ISPAD Clinical Practice Consensus Guidelines. Although this recommendation has little scientific evidence, the Japanese Society of Pediatric Endocrinology has adopted this value as the target HbA1c. We added the following text in support of the rationale for using these reference values:

“The Japanese Society for Pediatric Endocrinology (JSPE) translated these guidelines into Japanese in 2008. (6) The JSPE and the Japan Diabetes Society (JDS) recommended an HbA1c levels of < 7.5% for optimal glycaemic control since 2011, consistent with the ISPAD recommendations.(7) The Japanese Study Group of Insulin Therapy for Childhood and Adolescent Diabetes (JSGIT) uses this target in Japanese children and adolescents with T1DM.” (Line 59)

3. Please think about using graph to visualize the results.

Response: Thank you for your valuable advice. We presented some of the results as Figure 1, 2 and 3:

Fig. 1 Prevalence of obesity and overweight in each cohort

Fig. 2 Secular changes in total daily insulin dose, proportion of obesity, height SDSs and BMI percentile according to the level of glycaemic control in Japanese children with type 1 diabetes mellitus.

Fig. 3 Secular changes in height SDSs and BMI percentile in the four age groups.

4. Please rewrite the first paragraph of discussion, the audience does not need to read the table to know the key message of this study.

Response: Thank you very much for your comment. We deleted Tables in the first paragraph of the discussion, in accordance with your suggestion. (Line 203)

Response to Journal Requirements

1. We note that you have indicated that data from this study are available upon request. PLOS only allows data to be available upon request if there are legal or ethical restrictions on sharing data publicly.

Response: We added as follows, “Data cannot be shared publicly because data contain sensitive patient information. Data are available on request from the Japanese Study Group of Insulin Therapy for Childhood and Adolescent Diabetes and Ethics Committee of Saitama Medical University (contact via rinri@saitama-med.ac.jp) for researchers who meet the criteria for access to confidential data.”

2. One of the noted authors is a group, JSGIT. In addition to naming the author group and listing the individual authors and affiliations within this group in the acknowledgments section of your manuscript, please also indicate clearly a lead author for this group along with a contact email address.

Response: Thank you for your advice. As you pointed out, this study was performed by the JSGIT. This study was led by M. Mochizuki, and all co-authors were involved in the data analysis and writing the manuscript. We added the names, affiliations and email addresses of these group members to the acknowledgment section.

We hope we have addressed the main points raised by you and the reviewers. Once again, we are extremely grateful for your and the reviewers’ insightful suggestions and hope that the manuscript is now suitable for publication in PLoS One.

Sincerely,

Authors

*Corresponding author 

Mie Mochizuki

Department of Paediatrics, University of Yamanashi

1110 Shimokato, Chuo City, Yamanashi, Japan

Phone No: +81-(0)55-273-9606

Fax No: +81-(0)55-273-6745

Email Address: mmie@sweet.ocn.ne.jp

---

## [Decision Letter · Decision Letter 1]

7 Oct 2020

PONE-D-20-12969R1

Increasing secular trends in height and obesity in children with type 1 diabetes: JSGIT Cohort

PLOS ONE

Dear Dr. Mochizuki,

Thank you for submitting us your revison. One of our reviewers has a few relatively minor comments. Therefore, we hope very much that you will be willing and able to submit a revised version of the manuscript that addresses the points raised during the review process.

We look forward to receiving your revised manuscript.

Kind regards,

Yongfu Yu, Ph.D

Academic Editor

PLOS ONE

Reviewers' comments:

Reviewer's Responses to Questions

**Comments to the Author**

1. If the authors have adequately addressed your comments raised in a previous round of review and you feel that this manuscript is now acceptable for publication, you may indicate that here to bypass the “Comments to the Author” section, enter your conflict of interest statement in the “Confidential to Editor” section, and submit your "Accept" recommendation.

Reviewer #1: (No Response)

Reviewer #2: All comments have been addressed

Reviewer #3: All comments have been addressed

2. Is the manuscript technically sound, and do the data support the conclusions?

Reviewer #1: Yes

Reviewer #2: Yes

Reviewer #3: Yes

3. Has the statistical analysis been performed appropriately and rigorously? 

Reviewer #1: Yes

Reviewer #2: Yes

Reviewer #3: Yes

4. Have the authors made all data underlying the findings in their manuscript fully available?

Reviewer #1: No

Reviewer #2: No

Reviewer #3: Yes

5. Is the manuscript presented in an intelligible fashion and written in standard English?

Reviewer #1: Yes

Reviewer #2: Yes

Reviewer #3: Yes

6. Review Comments to the Author

Reviewer #1: Thank you for giving me an opportunity to review this article. The authors sufficiently responded my comments.

Minor comments I still have are the followings:

1. Line 244: "the increase that we observed in the use of bolus insulin at tea-time suggests that a subset of the patients had excess caloric intake". This is a strong statement without supporting data. Please rephrasing the sentence, at least adding "might" or similar to soften the argument.

2. Table 1 (number of participants). I may consider describing the p-value of the number of participants rather than just dichotomizing p-value (significant vs non-significant).

3. Table 2. Why are we missing p-for-trend in total daily insulin use? Also, how did the authors calculate p-for-trend in proportion of patients using insulin analogue? (e.g., at basal, did they calculate it using the data from 2008 and 2013? or did they also include "0" value in 1995 and 2000?) I would suggest adding the detailed explanation at least in the footnote of this table.

4. Figure 2. I appreciate the authors creating this nice figure. Please fix the x-label for c) SDS because it doesn't seem be in the appropriate position.

Reviewer #2: Thank you for the answers and corrections of the manuscript including addition of new Tables. My comments are considered to be addressed properly.

Reviewer #3: Thank you for your prompt reply. This paper is very impressive and important study to explore increasing secular trends in height and obesity in children with type I diabetes.

7. PLOS authors have the option to publish the peer review history of their article (what does this mean?). If published, this will include your full peer review and any attached files.

Reviewer #1: No

Reviewer #2: No

Reviewer #3: No

---

## [Author Response · Author response to Decision Letter 1]

28 Oct 2020

Responses to the Reviewer 

We thank the reviewers for their valuable comments and questions and for the opportunity to improve our paper. 

As requested, we prepared a revised version of our manuscript and hope that we have addressed the concerns of Reviewer #1. Our point-by-point responses to the reviewer’s comments follow, in this letter. 

We sincerely thank the reviewers for all of the helpful comments.

Responses to the comments of reviewer #1

1. Line 244: "the increase that we observed in the use of bolus insulin at tea-time suggests that a subset of the patients had excess caloric intake". This is a strong statement without supporting data. Please rephrasing the sentence, at least adding "might" or similar to soften the argument.

Response: We appreciate your insightful comment. We revised manuscript as follows:

“Although we did not assess patients’ diets in our study, the increase that we observed in the use of bolus insulin at tea-time might suggest that a subset of the patients had excess caloric intake.” (line 257)

2. Table 1 (number of participants). I may consider describing the p-value of the number of participants rather than just dichotomizing p-value (significant vs non-significant).

Response: We appreciate this recommendation to clarify the data in Table 1. The number of participants showed a significantly increased trend (P<0.001), but the proportion of boys did not show no significant trend. We revised Table 1, as follows:

3. Table 2. Why are we missing p-for-trend in total daily insulin use? Also, how did the authors calculate p-for-trend in proportion of patients using insulin analogue? (e.g., at basal, did they calculate it using the data from 2008 and 2013? or did they also include "0" value in 1995 and 2000?) I would suggest adding the detailed explanation at least in the footnote of this table.

Response: We appreciate your valuable suggestions to clarify these details. 

Total daily insulin dose showed no significant trend (P=0.992). Therefore, we determined P values <0.001 by the Mann–Whitney U test and compared the results with those in 1995. We revised the discussion of the findings in Table 2, as follows:

Because “in Japan, rapid-acting insulin analogues were introduced in 2000, and long-acting insulin analogues were introduced in 2003” (line 250), no patients used insulin analogues or insulin analogues as a bolus or as basal therapy. P-values were calculated using the data from 1995 to 2013, including a "0" value in 1995 or 2000. We added the following detailed explanation in the footnote to Table 1: “0, no patients used insulin analogues or insulin analogues as a bolus or as basal therapy.”, and “P-values for the trends as determined by the Cochran–Armitage or Jonckheere–Terpstra tests using data the from 1995 to 2013.”

4. Figure 2. I appreciate the authors creating this nice figure. Please fix the x-label for c) SDS because it doesn't seem be in the appropriate position.

5. 

Response: We appreciate your helpful comment. We moved the x-axis label to the appropriate position, as follows:

We hope we have addressed the main points raised by you and the reviewers. Once again, we are extremely grateful for your and the reviewers’ insightful suggestions and hope that the manuscript is now suitable for publication in PLoS One.

Sincerely,

Authors

---

## [Editor Report · Decision Letter 2]

30 Oct 2020

Increasing secular trends in height and obesity in children with type 1 diabetes: JSGIT Cohort

PONE-D-20-12969R2

Dear Dr. Mochizuki,

We’re pleased to inform you that your manuscript has been judged scientifically suitable for publication and will be formally accepted for publication once it meets all outstanding technical requirements.

Kind regards,

Yongfu Yu, Ph.D

Academic Editor

PLOS ONE
---

## [Editor Report · Acceptance letter]

12 Nov 2020

PONE-D-20-12969R2 

Increasing secular trends in height and obesity in children with type 1 diabetes: JSGIT Cohort 

Dear Dr. Mochizuki:

I'm pleased to inform you that your manuscript has been deemed suitable for publication in PLOS ONE. Congratulations! Your manuscript is now with our production department. 

Kind regards, 

on behalf of

Dr. Yongfu Yu 

Academic Editor

PLOS ONE